# Patients want to be seen: The top 3 information needs of patients with inguinal hernia

Karlijn J. van Stralen[1]*, Lotte Ruijter[1], Judith Frissen[1], Roeland H. den Boer[2], Veerle M. D. Struben[1], Catharina J. van Oostveen[1,3]

1 Spaarne Gasthuis Academy, Spaarne Gasthuis Hospital, Hoofddorp, Netherlands, 2 Surgical Department, Spaarne Gasthuis Hospital, Hoofddorp, Netherlands, 3 Erasmus School for Health Policy & Management, Erasmus University Rotterdam, Rotterdam, Netherlands

* KvanStralen@spaarnegasthuis.nl

**Data Availability Statement:** Data cannot be shared publicly because of privacy policy. Data are available from the Spaarne Gasthuis Local Ethical Committee (contact via ACLU@spaarnegasthuis.nl)

## Abstract

### Background

Good patient information has shown to improve surgical outcomes. In this study we explore what kind of pre-surgical information patients need and if the provision of a 360˚ video of a surgical procedure can be of added value to the information provided by the hospital.

### Methods

An explorative qualitative study using semi-structured interviews on information needs was conducted among 17 inguinal hernia patients to gain more insight in the patients' present surgical information needs. Patients either were planned to receive or already had received a surgical procedure. Questions were asked about the current information provision and, after being shown a 360˚ video of the surgery, whether this would be of added value.

### Results

Of the total group of 17 patients (mean age 56, interquartile range 45–64) 16 were male and one was female. Most had no previous experience with virtual reality (14/17), already had undergone a surgical procedure (11/17). Patient information needs were all about "seeing" which can be viewed from three different perspectives [1] being seen as a unique person in the treatment process, [2] being seen as a partner, and [3] seeing is understanding. Patients wanted the contact with the doctor to be more personal, with the possibility to see the anesthetist in person, the surgeon to see their wound in the recovery phase, and to receive personal answers to questions about their specific situation. Patients found the 360-video not fearsome, and believed that visual content could be beneficial as it appeals more to their imagination than written or oral information and increases their understanding. It also provided them with a better understanding of their treatment options, their pre-, peri-, and post-surgical procedures and identification of the cause of post-operative side effects.

for researchers who meet the criteria for access to confidential data.

**Funding:** This study was funded by the Spaarne Gasthuis innovation fund. The funders had no role in study design, data collection and analysis, decision to publish, or preparation of the manuscript.

**Competing interests:** The authors have declared that no competing interests exist.

## Conclusion

To address patients' information needs, complementary tools or services are needed that increase personal contact as well as tailor it to individual patient's needs. Even though video-apps are a partial alternative, hospitals should still offer patients the possibility of having face-to-face meetings with physicians as this is highly valued by patients and leads to increased trust in physicians' performance.

## Introduction

Research suggests that adequate information provision can reduce preoperative anxiety and may be beneficial for postoperative outcomes such as pain, behavioral recovery and the length of hospital stay [1, 2]. Furthermore, the information discussed in consultations forms the basis for informed consent for treatment, and patients look to clinicians to fulfil information needs [2, 3].

However, it is not always clear what information patients want as some prefer receiving very general information about just the procedure [4, 5], whilst others prefer detailed information on, for example, outcomes and survival [4, 6]. On the one hand, presenting large amounts of information may result in 'information overload' and extended pre-operative consultations, while on the other hand, just answering patients questions is only possible if patients have sufficient baseline knowledge [4].

Technological developments in the last decades have changed information provision. Most patients have access to the Internet, which offers unlimited amounts of health information [7]. In addition, many modern technology tools such as videos and e-health tools including apps and virtual reality have entered the health care market [1, 8, 9]. If patients are better informed this could impact information provision. For example, virtual reality (VR) offers patients the opportunity to watch videos in a different form; viewers are able to experience specific environments or situations which are not available in real-life, such as the sights and sounds of receiving and recovering from an anesthetic procedure [10]. Virtual reality has been used to train physicians or residents to improve their communication skills towards the patient [11]. Furthermore, one study suggested that informing patients on a surgical procedure via VR might improve patient comprehension of their condition [12].

In our hospital we had the impression that patients were missing certain information and that they did not read or (fully) understand the provided (written) information. This is important as during their meeting with their surgeon, patients are offered two options for surgery, as well have the option to postpone their procedure, and being well-informed beforehand may affect the patients choices [13]. Therefore, we aimed to explore what kind of pre-surgical information patients need, if modern electronic information tools could be an added value to current information provision, and if this information could influence their choices on their treatment.

## Methodology

This study was conducted and described according to the COnsolidated criteria for REporting Qualitative research (COREQ) checklist [14].

### Study design

We conducted an explorative qualitative study using semi-structured interviews to collect data on information needs of inguinal hernia patients at the surgical department of the Spaarne

Gasthuis hospital—a 400—bed teaching hospital -, Hoofddorp, The Netherlands. Interviews were held between September and December 2018. A laparoscopic hernia repair was chosen as the targeted surgical procedure as it is a often conducted surgical procedure and suitable and relevant for exploring patient's information needs.

## Participants

We selected participants using the purposive sampling technique. Only patients were included that had undergone a laparoscopic hernia repair during the preceding four weeks or that were scheduled to undergo this surgical procedure and already received all the pre-surgical information. A member of the surgical department approached patients by phone and after consent the interviewing researcher contacted them to plan the interviews. Interviews in Dutch were conducted at a time and place that patients preferred and all patients were interviewed once.

## Data collection

We collected data using semi-structured interviews, with the use of probes and prompts as interviewing technique. The interviews were audio recorded with the patients' permission and were guided by a topic list and consisted of two parts (Table 1). In the first part, the patient journey was used as a guide to direct the interview. Patients were asked about their information needs, the received information from the hospital and their information search behavior during the different stages of the health care process (from the first appointment with a general practitioner until discharge from the hospital and the post-operative period). In the second part of the interview, patients watched a 360˚ video with the VR headset. In this 360˚ video patients could virtually experience parts of the day of the surgery and see parts of a laparoscopic inguinal hernia repair procedure. Specifically, it showed scenes of patient's arrival in the holding area, the introduction of the surgical team, the anesthesia procedure, the transportation to the operating room, and some scenes of the laparoscopic surgery: the inguinal hernia, the insertion of the mesh and images of the repaired hernia (including the mesh). Voice-over was used to explain the procedure and the video had a length of 3.20 minutes. After having seen the video, patients were asked about their opinion of the video content, the VR headset and if this video was perceived of added value to the current pre-surgical information provision. The data collection was ended after saturation was reached.

**Table 1. Topic list interview.**

| | Main topic | Sub-topic |
|---|---|---|
| 1. | Diagnostic consultation | Preparation prior to consultation |
| | | Experience of the consultation |
| | | Needs regarding the consultation |
| 2. | Period between diagnostic consultation and date of surgery | Experience of this period |
| | | Activities during this period |
| | | Mental state during this period |
| | | Needs during this period |
| 3. | Impression of the VR-video | Experience of this video |
| | | Opinion of the design and content of the video |
| | | Opinion regarding the potential added value of the video |
| | | Preferences regarding form: 2d or 3d |
| | | Opinion regarding the user convenience (e.g. VR-headset) |
| | | Opinion about when in the treatment process this video should be presented |

### Research team and reflexivity

LR, a female researcher with a master's degree in clinical psychology conducted the interviews. KS and CO, both experienced researchers and responsible for study conduct and data analysis. JF is a junior researcher who assisted in the data-analysis. VS is member of the innovation board and together with RB (surgeon) responsible for the VR application. No relationship between the researchers and the patients was established before the study commencement. Patients were informed that the researchers were employed as researchers at the hospital where the patients were under treatment. Further information about the researchers was unknown to the patients.

### Data analysis

The recorded interviews were transcribed verbatim and thematically analyzed in Dutch. Data analysis was performed by coding both during and after the interviewing process, using MAXQDA software. Although both inductive and deductive coding was used to identify themes within the data, the analysis was more bottom-up than top-down. The analysis was based on the six-phase framework for doing a thematic analysis developed by Braun and Clarke [15]. During the first phase of the coding process interviews were transcribed and read by LR, CO and JF to become familiar with the data and identify possible interesting codes. During the second phase, initial codes were generated based on the data itself by using patients' words for codes (inductive). The third phase involved searching for themes by grouping and linking the data-driven codes. In the subsequent phase, grouped codes were reviewed and renamed by using a more deductive approach as we wanted to focus on identifying patients' information needs. The fifth phase included defining the themes and the last phase was used for reporting the themes. The interviews were coded by LR and reviewed by CO and JF to reach consensus about the coding tree, e.g. the codes used and the way they were grouped and linked.

### Ethical consideration

The local research ethics committee of the hospital (advies commissie lokale uitvoerbaarheid —ACLU) evaluated the ethical acceptability of this study and gave approval. Furthermore, each participant gave written informed consent at the start of the study. Additionally, all data were treated as confidential and participant's anonymity was warranted by dissociation of names. Data were stored under identification numbers, which were randomly designated and conserved by one of the researchers (CO).

## Results

In total, 21 patients were contacted and 17 patients participated in the study. One of the four patients that refused to participate found it too stressful to take part as this patient still had to undergo the surgery. Another patient who still had to undergo the surgery had mental issues and therefore his wife preferred that this patient would not participate in the research. The last two patients refused to participate due to upcoming exams and time constraints. The mean age of the participating patients is 56 years (SD: 17; min. 35, max. 79) and 16 were men (94%). Eleven of the seventeen patients (64.7%) had undergone surgery while six were scheduled to get surgery. Interviews lasted 40 to 60 minutes and were held in the home setting (n = 5), hospital (n = 8), at work (n = 3) and one took place by phone (this patient was not interested in the video or any other information about the surgical procedure apart from the time and

date). During three of the interviews the partner of the participant was present while otherwise only the participant and interviewing researcher were present.

During the coding and analyses phases, it became clear that most of the patient remarks were referring to an overall theme of "seeing". Apparently, a visual tool helped them to be seen as a unique person and partner in the treatment process. Here, 'being seen as a unique person' reflects a more personal (authentic) connection with the healthcare providers. 'Being seen as a partner' refers to the value of being viewed as a partner or equal in the treatment process. The actual use of visual tools resulted in the third theme "seeing is understanding", which provides insights into whether visual tools help patients to better understand pre-surgical information and getting involved in the treatment process (Table 2).

### Being seen as a unique person

Several patients emphasized the importance of personal contact, either in person or via visual tools. This gave patients the feeling of being seen as an authentic person and not as a number as it also provided them with more tailored advice.

**In person.** During the preoperative phase, after patients decided to opt for surgery based on a personal surgical assessment, patients have an appointment with the anesthesiologist. During this consultation, information is provided about the different methods of anesthesia and the behavior requirements before surgery. In several cases, this appointment took place by phone. Various patients stated they (had) preferred to go to the hospital for this consultation, as it made them feel treated as an individual and not as one of the hundreds of patients. For instance, patients mentioned that seeing the anesthesiologist enabled the anesthesiologist to better determine the patient's individual needs and to adjust the treatment process according to these needs. Moreover, patients indicated that seeing the face of the anesthesiologist in person brought them trust in his or her expertise.

> "They asked questions to ensure the process would fit my needs. They checked my pulse and whether I have loose teeth or prostheses etcetera, so they ensured they would act safely later on. We also decided that I would get a light tranquilizer on my arrival at the hospital because I was very worried about the fact that when you get anesthesia you stop breathing, and they take over the breathing process."–P3

In addition, patients found it important to be seen by the surgeon in the postoperative phase because they wanted reassurance about the recovery process. Nowadays this is only done by phone. Patients mentioned that they found it difficult to judge their own recovery process and therefore preferred to show their wound to the surgeon (either in person or via video apps) and not reporting about the wound via phone.

**Table 2.**

| (*Sub*)Theme | Definition |
|---|---|
| Being seen as a unique person | Having a personal (authentic) connection with the healthcare providers |
| *Being seen in person* | *Being seen physically by the physician and not via the telephone* |
| *Use of tools* | *Being informed through online contact via messaging apps* |
| Being seen as a partner | Being viewed as a partner or equal in the treatment process to allow for shared or informed decision making |
| Seeing is understanding | Use of visual tools to help understand pre-surgical information |

*"I was surprised to receive a phone call regarding my state after the surgery. I thought to myself it would be better if this was by FaceTime because then I could show the doctor what the wound looked like."–P11*

**Using tools.** When patients were informed or kept up to date through online contact, they had the feeling they were taken more seriously in the treatment process. Some patients stated they were in touch with their surgeon through messaging apps during the recovery process. Patients mentioned that online contact during the trajectory could provide the opportunity to get more tailored advice.

*"So I sent a [WhatsApp] message to the surgeon... 'I would like to know how long it takes before I am able to go back to school and how much rest I really should take'. The surgeon told me I should be conservative the first week and after that I could restart things if it feels ok. The only thing I couldn't do yet was carry heavy things. The information I received from the surgeon was the information I specifically needed and was clearer than the information I got from the outpatient department."–P1*

Furthermore, various patients indicated that it was necessary to be informed earlier about the date of surgery. Patients sometimes needed to wait for months until they were offered an appointment and only knew the exact operation date shortly in advance. As a result, patients found it difficult to plan their personal activities due to a lack of an indication of the operation date. One patient suggested that the waiting list could be posted online with a personal login code. This way the patient could have a better indication of when the surgery will take place.

*"Patients should receive the same kind of notifications as they get from the dentist, there you also receive a notification like 'hello, you need to go to the dentist tomorrow'<...> That you at least receive an update about the status."–P13*

Another patient suggested that on the day of surgery it would be useful if the family of the patient receives an SMS when the patient has awakened after the surgery procedure.

*"It should be possible to tell in phases what is going to happen and at which moment, for example.. that the family members or contact person receive an SMS when the patient, daughter or family member, is ready when the surgery procedure is finished, and he or she lies in the recovery room"–P11*

## Being seen as a partner in the treatment process

The majority of the interviews with patients outlined the necessity of a (perceived) control over the treatment process and active involvement in decision-making.

Patients who had informed themselves via, for example, the internet perceived themselves as a more active and equal partner in the treatment process than those who had not. Half of the patients had searched for information on the Internet; some patients stated that they used the Internet to check if the given diagnosis and advised treatment were correct. Other patients just wanted more information, about what an inguinal hernia is, the treatment options, possible complications and anesthesia. Some patients even watched surgical procedures on YouTube.

*"I had a look on the internet to see if it actually could be that I had an inguinal hernia, so whether the symptoms were matching. I was also a bit curious about what options there were to recover from it."–P15*

For others, the Internet was not perceived as helpful in providing sufficient confidence for an active role in the shared decision-making process. These patients experienced an information overload on the Internet or too much complexity within the information. Hence, some patients found it difficult to filter information and to draw the right conclusions. A few patients believed that there was too much misinformation or exaggerated information on the Internet. For some patients this was the reason to not search further for information on the Internet.

*"I looked a bit on the Internet to see what the consequences of the surgery are, but you can better not look on the Internet as only negative experiences have been posted online and not positive ones. I also had a look at the different treatments, but I could not get any wiser from that."–P7*

After being provided with the 360˚ video some patients indicated that it helped them in asking relevant questions. This gave them the feeling that they had enough understanding to have a valuable dialogue with the physician. Therefore, most of the patients indicated that the best moment for presenting this 360˚ video would be just before the first consultation in the hospital, so that they would be able to address this information during the consultation.

*"Hospitals need to be more flexible, transparent and I think this is a very good step towards it. You know, this contributes to that and in the past it used to be like.. you need to go the hospital, you undergo surgery and you know, what we say.. take it or leave it, you know, that's how it is. And with this sort of things you feel taken seriously as a patient, and we want to be more informed these days and this [the video] helps."–P3*

However, a few patients stated that they only needed information about the date of surgery and recovery time. These patients did not want much information because they needed to undergo surgery anyway; they did not see it as a choice and just wanted it to be over.

*"For me it is just important to know how long this thing will last and when I will be able to get back to work and feel good again, these kinds of things. This was the most important to know. How the surgery procedure is performed exactly and how it works, as far as I understand, a mesh is inserted. How it looks like, how it works etc. to be honest, it does not really interest me. As long as my problem is solved, I am fine".–P2*

The pre-operative phase is also a phase in which a patient must choose between various options with respect to the surgery. In this particular case, there were two different options with regard to the type of surgery, two different options with respect to anesthesia, and sometimes they could choose not to undergo the surgery or to postpone it. In some cases only one option was suitable due to certain circumstances, whereas in other cases more than one option was suitable.

*"Oh, so I had the endoscopic surgery.. ooh."–P14*

*"I wanted to know how the anesthesia would go. Well.. sometimes you will get the spinal.. I did not know, you have a spinal block and I do not have a clue what else. I think they can also inject something, somewhere else in an infusion. . . or something with nitrous oxide, I also did not know so well how that was going. The anesthesiologist said that I do not have to worry."– P1*

Only a few patients were actively involved in the decision-making regarding the options for the type of surgery or the form of anesthesia, or whether to undergo the surgery.

Sometimes this was because the physician provided them with the choice, but mostly this was because the patient had a different preference due to previous experience with an inguinal hernia surgery repair.

*"The doctor explained it could be a spinal block or a complete anesthesia. I will just have general anesthesia as I find that more comfortable."–P4*

One of the patients suggested that the hospital could share experiences from other patients, so that it would become easier to make a decision and patients would know what they should or should not do.

*"Well, now we are talking.. I think.. this surgery is a surgery that happens a lot. They told that when we entered the department. Then you should ask the people who had undergone that surgery this year.. well that might be too much people, but if you would ask 100 people about their experiences, than people could benefit from this information and know when a choice was not so bad".–P11*

Noteworthy is that some patients who had not yet undergone the surgery, were thinking about postponing the surgery after seeing the 360˚ video, as the video made them realize what the impact of the surgery would be. As a result, this made them think about which period would be a suitable period for them to undergo the surgery. This implies that the video triggered some involvement in decision-making.

*"I was thinking maybe I should postpone it until after summer.. now I see how long it could take before I fully will function again after surgery..".–P14*

## Seeing is understanding

Several patients experienced a need for additional information and believed (or were under the impression) that visual content could be beneficial as it appeals more to their imagination than written or oral information and therefore increases their understanding. Visual content proved to be especially suitable to provide procedural information pre-, peri- and post-surgical.

The majority of the patients did not know what would happen between the moment they would arrive at the hospital for the surgery and the moment they would leave the hospital again. Patients noted that the 360˚ video corrected this lack of information. Moreover, patients expressed that the provision of this information in written or oral form would not have been as effective as visual content. Patients stated that the visual content in the 360˚ video increased their trust as they could see that the hospital would really take care of them before and during the surgery. Also, they indicated that they would be more at ease on the day of surgery as they better knew what to expect.

*"You are able to see something which you are not going to see later on because you will be unconscious.. but you know in advance you're going to a kind of familiar environment.. you have already seen it once so you can.. it makes me feel relaxed. Oh, this is how it looks like.. someone is monitoring my heartbeat continuously and you know. . . my physical condition. There's someone who's just monitoring, well fine, I am happy with that. Good to know, I didn't know."–P3*

However, some patients wondered if the hospital should always inform people about the surgical procedure by showing real procedure images.

*"If you have a huge tumor in your belly. . . it is an operation of 12 hours, should you use such a video to inform patients? This is what I am asking myself.. Look, this [inguinal hernia repair] is a simple surgical procedure and some people might want to know how it works, well, put the video on the internet with a link and patients are able to watch it if they want to."–P6*

Various patients that already had undergone the surgery noted that seeing the 360˚ video helped them to understand some post-operative outcomes. For instance, because of seeing the 360˚ video, a handful of patients came to realize why they had abdominal distension after surgery. The reason for this is that during surgery, the patient's abdomen will be filled with gas to identify the weak spot in the abdominal wall.

*"Gas.. the gas I did not know.. the gas. When I came home after surgery, I had the feeling that I had a big belly although I am someone who, in general, is watching my figure. I thought why.. why on earth did I get this big belly? But now I see that it is because of the gas they put into my belly."–P5*

Other patients only realized after seeing the 360˚ video why they had felt the pain at a particular location of their body. They did not know that that was the spot where the surgical mesh was placed as they thought it was placed elsewhere. Some patients finally understood some complications that they were concerned about, and they would have appreciated seeing this 360˚ before surgery.

*"I really felt pain in my shoulder during the night and I did not like that because I did not know it was normal to feel pain in my shoulder after the surgery. Yes, it happened to be that a friend of mine knew it was one of the side effects of an inguinal hernia repair and now in this video they let you know, so that is very good.".–P10*

After watching the 360 video, several patients mentioned that the VR headset gave them the feeling that they were really inside the operation room. They really felt present. Most of the patients found the VR glasses entertaining and fun to use and wanted to be provided with information through VR more often.

*"I don't think you can get this effect on a normal television screen. Now you can.. it is like you are really there. It is an experience."—P12*

However, some did not find it necessary to experience the surgery in such a way. They mentioned that looking at an animation video at a 2D screen would be sufficient for them.

*"If it just had been an animation video, it would also fine to me.. or just a doctor explaining it to me without the real images of the inside of the belly, but just highlighting with a pencil the part of the belly which they will enter and where they will place the mesh".–P10*

Still, the use of the VR-video was accepted by all patients and none of the patients found it fearsome themselves. Beforehand we had expected that patients might find it fearsome to view a surgical procedure in 360. Some patients indicated that it could scare people off to get so close to the real surgery, although they did not find it problematic themselves. In addition,

there was one patient who indicated not to be interested in the video and who had the interview by phone.

## Discussion

This study explored the pre-surgical information needs of patients nowadays and investigated whether the provision of a 360° video can be of added value to the information that is currently being provided by the hospital. The results revealed the main theme 'seeing' with respect to information needs of patients, which can be viewed from three different perspectives [1] Being seen as a unique person and [2] being seen as a partner, and [3] Seeing is understanding. Patients wanted the contact with the doctor to be more personal, with the possibility to see the anesthetist in person, the surgeon to see their wound in the recovery phase, and to receive clear answers to questions regarding their specific personal situation. Experiencing the surgical procedure by VR helped them in understanding treatment options, pre- peri- and post-surgical procedures and post-surgical side effects.

### Being seen as a unique person and partner

Patients indicated that they wanted to be seen as a unique person in the treatment process. Currently, healthcare changes are focused on making processes more efficient. Even though research has shown that a preoperative evaluation and post-operative consultations can be safely done by phone in many cases, some patients indicated this made them feel like a number instead of a person. This is in contrast with the literature, showing a high satisfaction using telemedicine, with scores ranging from 4.5 to 5 out of 5 [16]. However, most of the studies included in this review were conducted before 2010. Since then, video apps have become widely common in everyday life and patients have gotten used to seeing their callers and their surroundings. Therefore, opinions with respect to contact by "normal" phone may have changed. Patients indicated that they wanted to show their wounds to the surgeon, instead of just reporting on it. Other studies have shown great effectiveness and satisfaction with video-calling which may increase the satisfaction, instead of just contact by phone [16].

Patients felt that the general information obtained from the brochures, the internet or doctors assistants, was not sufficiently tailored to their personal situation, for instance for questions regarding their recovery process. It had been shown previously that tailored information provided on the internet leads to higher satisfaction rates among patients, especially among younger patients, although no improvements in patients outcomes have been found [17, 18]. Some of our patients had WhatsApp contact with their surgeon, which was highly appreciated. However, this would be nearly impossible to provide to all patients, as physicians are often afraid of being overloaded with questions from patients. Still research has shown benefits of a short messaging system protocol leading to a reduced number of clinic visits and increased efficiency of clinic visits [19]. Alternative options could be for example use of chatbots. For the latter, studies in breast cancer patients, for example, have shown that an artificial conversation agent achieved equal satisfaction scores as "real" physicians [20].

Patients were not always a partner in the decision-making process concerning their treatment as they indicated that they were not aware of the treatment they had received and that there were multiple options. Some even indicated that they intended to postpone their surgery after seeing the video. They said they would have liked to see the video shortly before their pre-operative consultation, so they would be able to memorize all the questions. Patient's involvement in decision-making is only lately becoming more common and as a result more and more decision-making interventions and interventions that promote patient's expertise are being developed [21]. A procedural 360° video might be one of these tools and such

interventions enable patients to gain the right knowledge to negotiate with the physician about how the best treatment process would look like.

## Seeing is understanding

The majority of the patients believed that the 360˚ video enhanced their understanding, especially with regard to the surgical procedural information. Retrieving procedural information is crucial for proper psychological preparation prior to surgery [22] via for instance less surprise during the day of surgery [10]. In turn, the psychological preparation will influence the outcomes of the surgery as the way people think and feel have impact on the operation outcomes [2]. Patients noted that seeing the procedure increased their trust and familiarity with the operation environment, thereby decreasing their fear. Seeing is beneficial as it was found that diminishing the patient's fear before surgery results in patients experiencing higher quality of the treatment in all its different phases [1].

Patients also indicated that it improved their knowledge on the post-operative outcomes. Patients, who already had undergone surgery, said that seeing the 360˚ video made them realize why they had certain side effects after surgery such as abdominal distension or pain in the shoulder. In case a patient expects to experience some soreness in his or her shoulder after surgery, the actual experience of soreness in the shoulder is seen as a normal side effect, which in turn diminishes the feeling that something went wrong. This is crucial, as catastrophizing would have been likely to increase the pain [23]. Also in other fields it has been suggested that VR can help in pharmacy education by providing immersive educational activities [24], having elderly patients coping with dementia [25] or teaching skills to health professionals [26].

Multiple patients expressed that the video helped them for instance to ask relevant questions during the consultation with the physician, which in turn is likely to promote informed decision-making. Furthermore, the patients might feel more comfortable in contact with the surgeon referring to a video (or another tool) provided by the hospital than provided by the Internet.

## Strengths and limitations

When interpreting this study and its results, some strengths and limitations must be considered. The strength of this study lies in the use of a 360˚ video in a VR device to prepare adult patients for surgery. The majority of research on the use of VR in the clinical setting focuses on how this medium can contribute to physician's surgical practice and not on the potential added value this medium can have for patients. Some studies that did investigate how VR could contribute to the patient's preparation before surgery mainly focused on children rather than adults. As far as we know, the use of VR to prepare adult patients for a small surgical procedure has not yet been studied.

It should be addressed that it is unclear to what extent the results can be generalized to patients undergoing another surgical procedure than inguinal hernia surgical repair. Patients believe that it should be used more often for relatively simple procedures. Furthermore, this study only used a non-interactive form of VR in which the patients could look up, down, left, right and around, but could not walk forward or interact with the virtual objects. Hence, the potential of an interactive virtual environment for informing patients before surgery has not been studied, but might be more suitable for educational purposes than information provision. Although this is normal in this patient population, there was only one female patient. Finally, the patient population was relatively young (56 years) as compared to the average patient population, so this might have biased the results towards needs for personal contact. Still, most of the patients were unfamiliar with a VR device.

## Practical implications & conclusions

This study highlights the importance for hospitals to provide patients with modern tools such as a 360˚ video as to fulfill the information needs of the patients, and more specifically to enable patients' involvement in informed decision-making regarding the treatment process. When creating or selecting these tools, hospitals should consider the following: the ability for personal contact, opportunities for tailoring to individual needs, visual content, immersion and procedural knowledge to facilitate shared decision-making. The 360˚ video used in this study did fulfill some of these needs, such as triggering informed decision-making with regard to the timing of the surgery, or providing visual content and immersion with regard to both procedural- and sensorial information thereby filling the information gap on what to expect during and after the surgery.

The current COVID-19 outbreak has moved many consultations out of the hospitals. To address to patients' information needs, complementary tools or services are needed that increase personal contact as well as tailoring with respect to individual needs. Even though video-apps are a partial alternative, hospitals should still offer patients the possibility for face-to-face meetings with physicians as this is highly valued by patients and leads to increased trust in physicians' performance.

## Author Contributions

**Conceptualization:** Karlijn J. van Stralen, Roeland H. den Boer, Veerle M. D. Struben.

**Data curation:** Lotte Ruijter, Judith Frissen, Catharina J. van Oostveen.

**Formal analysis:** Lotte Ruijter, Catharina J. van Oostveen.

**Funding acquisition:** Karlijn J. van Stralen.

**Methodology:** Catharina J. van Oostveen.

**Supervision:** Karlijn J. van Stralen, Roeland H. den Boer, Veerle M. D. Struben, Catharina J. van Oostveen.

**Validation:** Karlijn J. van Stralen.

**Writing – original draft:** Karlijn J. van Stralen, Lotte Ruijter.

**Writing – review & editing:** Karlijn J. van Stralen, Judith Frissen, Roeland H. den Boer, Veerle M. D. Struben, Catharina J. van Oostveen.

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
