## [Decision Letter · Decision Letter 0]

17 Mar 2020

PONE-D-19-28443

Top 3 information needs of patients in 2020

A qualitative study among patients with inguinal hernia.

PLOS ONE

Dear Mrs van Stralen,

Thank you for submitting your manuscript to PLOS ONE. After careful consideration, we feel that it has merit but does not fully meet PLOS ONE’s publication criteria as it currently stands. Therefore, we invite you to submit a revised version of the manuscript that addresses the points raised during the review process.

I am sorry for the delay in processing this submission. I was not the original assigned editor. Please do give careful consideration to the reviewer's comments. I also have a query about language. I presume the interviews were originally done in Dutch, and have been translated into English. You should describe the translation process and how you tried to assure accuracy. I note that on two occasions, your interviewees are said to have referred to spinal anaesthesia calling it a "spinal tap." The correct term for spinal anaesthesia is a spinal block. (There are also more complicated terms for it.) "Spinal tap" refers to the extraction of cerebrospinal fluid for diagnostic purposes, a completely different procedure. If the interviews were done in English and the respondents did in fact say that, you should point out that they are in error. If these were translated, this is a mistranslation which leaves me uncertain as to the translation quality in general. The English in the text is generally readable, but at times stilted and there are some odd word choices. Please have this reviewed by a native or fully fluent English speaker. Also, the title refers to 2020 but this research was done in 2019 or earlier. You should delete the year from the title.

We would appreciate receiving your revised manuscript by May 01 2020 11:59PM. To enhance the reproducibility of your results, we recommend that if applicable you deposit your laboratory protocols in protocols.io, where a protocol can be assigned its own identifier (DOI) such that it can be cited independently in the future. For instructions see: http://journals.plos.org/plosone/s/submission-guidelines#loc-laboratory-protocols

We look forward to receiving your revised manuscript.

Kind regards,

M Barton Laws

Academic Editor

PLOS ONE

Journal Requirements:

"The local research ethics committee of the hospital evaluated the ethical acceptability of this study and gave approval.".

i) Please amend your current ethics statement to include the full name of the ethics committee/institutional review board(s) that approved your specific study.

ii) Once you have amended this/these statement(s) in the Methods section of the manuscript, please add the same text to the “Ethics Statement” field of the submission form (via “Edit Submission”).

"This study was funded by the Spaarne Gasthuis innovation fund. The funders had no role in study

design, data collection and analysis, decision to publish, or preparation of the manuscript. We have no

conflict of interest to declare.".

i) We note that you have provided funding information that is not currently declared in your Funding Statement. However, funding information should not appear in the Acknowledgments section or other areas of your manuscript. We will only publish funding information present in the Funding Statement section of the online submission form.

ii) Please remove any funding-related text from the manuscript and let us know how you would like to update your Funding Statement. Currently, your Funding Statement reads as follows:

"no; The funders had no role in study design, data collection and analysis, decision to publish, or preparation of the manuscript".

Additional Editor Comments (if provided):

I am sorry for the delay in processing this submission. I was not the original assigned editor. Please do give careful consideration to the reviewer's comments. I also have a query about language. I presume the interviews were originally done in Dutch, and have been translated into English. You should describe the translation process and how you tried to assure accuracy. I note that on two occasions, your interviewees are said to have referred to spinal anaesthesia calling it a "spinal tap." The correct term for spinal anaesthesia is a spinal block. (There are also more complicated terms for it.) "Spinal tap" refers to the extraction of cerebrospinal fluid for diagnostic purposes, a completely different procedure. If the interviews were done in English and the respondents did in fact say that, you should point out that they are in error. If these were translated, this is a mistranslation which leaves me uncertain as to the translation quality in general. The English in the text is generally readable, but at times stilted and there are some odd word choices. Please have this reviewed by a native or fully fluent English speaker. Also, the title refers to 2020 but this research was done in 2019 or earlier. You should delete the year from the title.

Reviewers' comments:

Reviewer's Responses to Questions

**Comments to the Author**

1. Is the manuscript technically sound, and do the data support the conclusions?

Reviewer #1: Yes

2. Has the statistical analysis been performed appropriately and rigorously? 

Reviewer #1: N/A

3. Have the authors made all data underlying the findings in their manuscript fully available?

Reviewer #1: Yes

4. Is the manuscript presented in an intelligible fashion and written in standard English?

Reviewer #1: Yes

5. Review Comments to the Author

Reviewer #1: PONE-D-19_28443

Top 3 information needs of patients in 2020 A qualitative study among patients with inguinal hernia.

General Comments

The authors interviewed patients who had or were scheduled for a hernia surgery on the use of a 360 video of their surgery. Given the scope of patient interest in being engaged in their treatment as well as growing support for technology to support patients’ needs, this study is both timely and unique. The study was simple in design and thoughtfully conducted. The findings reflect themes elaborating the value of the innovative nature of the intervention the utility of supporting patients in non-traditional ways to improve follow-up and experience.

Abstract:

Please include some brief patient demographics

Introduction:

The background is simply insufficient. What is the context for this work. What else has been done. Why is this work important in the field, and for patients and physicians?

More discussion around patients involvement in treatment would be useful.

Also, this is such an important area of innovation and thinking, the authors should talk at some depth around the advancement of technology in medical decision making and patient-centered care.

Method:

Some further elaboration on what “coding” is, and how the process of coding was completed. In other words, how did the authors get from “coded” elements in the transcripts to “themes”. How were codes prioritized, how was meaning ascribed to the codes, and in what ways did inductive vs deductive approach assist in the process (bottom up vs top down). Essentially, a bit more explanation here in the way of definitions, an example, or a description of an inductive vs deductive code would be useful for non-qualitative researchers. In that way, a definition or explanation of a “coding tree” versus a concept map, might be helpful.

Results:

There seem to be three main themes or groups of themes presented. The authors could organize their results a little more to give more life to these important findings. I will try to illustrate this need below.

The set of findings (themes 1 and 2) would benefit from a bit more description in the themes or in the description of themes. What I mean to say is that the themes all link to the “visual tool” and would be more descriptive if they could reflect that connection, i.e., “Visual tool helped patients be seen as unique and partner in treatment process.”

A little more organization within the themes (1 and 2) would help pull the findings together around the two main ideas - being seen (literally) and being seen (figuratively). While these are generally okay, they obfuscate the subtle and arguably more important differences between them. In my view, your argument about being seen (visually) reflects a more personal (authentic) connection with the provider. The other theme, being seen (figuratively) seems to reflect more of a value of being trusted or being viewed as a partner or equal in the treatment process. While I recognize that the authors’ use of literal and figurative are correct, I worry in some way that their close association makes it harder to see the valuable differences in the themes.

Then there is the third theme. The third theme, or category of themes, “Seeing is understanding” is very revealing and offers important insight into the patient experience. I worry that because the authors do not organize the results in a way to prepare the reader for three larger themes with sub-themes, this final section loses its punch.

The authors could elaborate a bit more using their own narrative between illustrative quotes. While very helpful, the quotes lose their strength when grouped together. In other words, subtlety in the quotes should be drawn out with the use of descriptive narrative. Or if the quotes seem to reflect the same idea, it would be better to pick one quote rather than use two or three to convey the same idea.

Given the work conducted by the authors to “identify, label, and code” the transcripts, it is hard to believe that the themes “emerged” from the transcripts. I believe the authors did lots of work to examine the text and make well-intentioned choices when they derived the final “themes”. The section would be improved if authors could use a verb other than “emerged” to describe how the “themes” were generated by a thorough process of coding, analysis, and achieving consensus.

6. PLOS authors have the option to publish the peer review history of their article (what does this mean?). If published, this will include your full peer review and any attached files.

Reviewer #1: No

---

## [Author Response · Author response to Decision Letter 0]

4 Aug 2020

Editor

I am sorry for the delay in processing this submission. I was not the original assigned editor. Please do give careful consideration to the reviewer's comments. I also have a query about language. I presume the interviews were originally done in Dutch, and have been translated into English. You should describe the translation process and how you tried to assure accuracy. I note that on two occasions, your interviewees are said to have referred to spinal anaesthesia calling it a "spinal tap." The correct term for spinal anaesthesia is a spinal block. (There are also more complicated terms for it.) "Spinal tap" refers to the extraction of cerebrospinal fluid for diagnostic purposes, a completely different procedure. If the interviews were done in English and the respondents did in fact say that, you should point out that they are in error. If these were translated, this is a mistranslation which leaves me uncertain as to the translation quality in general. The English in the text is generally readable, but at times stilted and there are some odd word choices. Please have this reviewed by a native or fully fluent English speaker. Also, the title refers to 2020 but this research was done in 2019 or earlier. You should delete the year from the title.

Thank you for your comments. We adapted the manuscript accordingly, changed the title, and had the manuscript checked by a native speaker. We thank both the editor and the reviewer for their comments as we believe that they greatly improved our paper. 

Reviewer 1

The authors interviewed patients who had or were scheduled for a hernia surgery on the use of a 360 video of their surgery. Given the scope of patient interest in being engaged in their treatment as well as growing support for technology to support patients’ needs, this study is both timely and unique. The study was simple in design and thoughtfully conducted. The findings reflect themes elaborating the value of the innovative nature of the intervention the utility of supporting patients in non-traditional ways to improve follow-up and experience.

We thank the reviewer for these encouraging comments, and the comments as we believe that they greatly improved our manuscript. 

Abstract: 

Please include some brief patient demographics

We added some patient demographics, also to the manuscript itself

Introduction:

The background is simply insufficient. What is the context for this work. What else has been done. Why is this work important in the field, and for patients and physicians?

More discussion around patients involvement in treatment would be useful. Also, this is such an important area of innovation and thinking, the authors should talk at some depth around the advancement of technology in medical decision making and patient-centered care.

We agree with the reviewer that the introduction was a bit (too) short and therefore rewrote the introduction. 

Method:

Some further elaboration on what “coding” is, and how the process of coding was completed. In other words, how did the authors get from “coded” elements in the transcripts to “themes”. How were codes prioritized, how was meaning ascribed to the codes, and in what ways did inductive vs deductive approach assist in the process (bottom up vs top down). Essentially, a bit more explanation here in the way of definitions, an example, or a description of an inductive vs deductive code would be useful for non-qualitative researchers. In that way, a definition or explanation of a “coding tree” versus a concept map, might be helpful.

We have added elaboration on the coding process by describing more specifically how the coding process was organized and how inductive and deductive coding was used. In addition, more explanation was given about the coding tree. 

Results:

There seem to be three main themes or groups of themes presented. The authors could organize their results a little more to give more life to these important findings. I will try to illustrate this need below.

seeing”

The set of findings (themes 1 and 2) would benefit from a bit more description in the themes or in the description of themes. What I mean to say is that the themes all link to the “visual tool” and would be more descriptive if they could reflect that connection, i.e., “Visual tool helped patients be seen as unique and partner in treatment process.”

A little more organization within the themes (1 and 2) would help pull the findings together around the two main ideas - being seen (literally) and being seen (figuratively). While these are generally okay, they obfuscate the subtle and arguably more important differences between them. In my view, your argument about being seen (visually) reflects a more personal (authentic) connection with the provider. The other theme, being seen (figuratively) seems to reflect more of a value of being trusted or being viewed as a partner or equal in the treatment process. While I recognize that the authors’ use of literal and figurative are correct, I worry in some way that their close association makes it harder to see the valuable differences in the themes.

Then there is the third theme. The third theme, or category of themes, “Seeing is understanding” is very revealing and offers important insight into the patient experience. I worry that because the authors do not organize the results in a way to prepare the reader for three larger themes with sub-themes, this final section loses its punch.

To put more emphasis on the third theme we introduced the three themes more clearly, and restructured all themes by renaming themes, removing quotes and rephrasing and shuffling paragraphs. 

The authors could elaborate a bit more using their own narrative between illustrative quotes. While very helpful, the quotes lose their strength when grouped together. In other words, subtlety in the quotes should be drawn out with the use of descriptive narrative. Or if the quotes seem to reflect the same idea, it would be better to pick one quote rather than use two or three to convey the same idea.

We carefully discussed all the quotes and checked which one were duplicates and removed them. 

Given the work conducted by the authors to “identify, label, and code” the transcripts, it is hard to believe that the themes “emerged” from the transcripts. I believe the authors did lots of work to examine the text and make well-intentioned choices when they derived the final “themes”. The section would be improved if authors could use a verb other than “emerged” to describe how the “themes” were generated by a thorough process of coding, analysis, and achieving consensus.

The reviewer is correct. During the coding and analyses phase it became clear that many patients literally used the word seeing (or being seen etc) many times and that many of the other factors were relating to this same ‘overall topic”. We therefore reprased the introduction of the result section. 

We hope that this revision will be followed by further reviewing/ publication of our article and are looking forward to hearing your response.

Yours sincerely, on behalf of all co-authors,

Karlijn van Stralen PhD

---

## [Decision Letter · Decision Letter 1]

7 Sep 2020

PONE-D-19-28443R1

Patients want to be seen: the top 3 information needs of patients with inguinal hernia.

PLOS ONE

Dear Dr. van Stralen,

Thank you for submitting your manuscript to PLOS ONE. After careful consideration, we feel that it has merit but does not fully meet PLOS ONE’s publication criteria as it currently stands. Therefore, we invite you to submit a revised version of the manuscript that addresses the points raised during the review process.

I apologize for the delay in getting a review of this submission. While the reviewer has asked for major revision, most of the comments have to do with style, presentation and completeness. Please do pay particular attention to consistency, copy editing and English style. If these issues are satisfactorily addressed, it is possible that another round of peer review will not be necessary.

We look forward to receiving your revised manuscript.

Kind regards,

M Barton Laws

Academic Editor

PLOS ONE

Reviewers' comments:

Reviewer's Responses to Questions

**Comments to the Author**

1. If the authors have adequately addressed your comments raised in a previous round of review and you feel that this manuscript is now acceptable for publication, you may indicate that here to bypass the “Comments to the Author” section, enter your conflict of interest statement in the “Confidential to Editor” section, and submit your "Accept" recommendation.

Reviewer #2: (No Response)

2. Is the manuscript technically sound, and do the data support the conclusions?

Reviewer #2: Yes

3. Has the statistical analysis been performed appropriately and rigorously? 

Reviewer #2: N/A

4. Have the authors made all data underlying the findings in their manuscript fully available?

Reviewer #2: (No Response)

5. Is the manuscript presented in an intelligible fashion and written in standard English?

Reviewer #2: Yes

6. Review Comments to the Author

Reviewer #2: The authors report their findings of qualitative interviews with patients who had or were going to have inguinal hernia repair surgery and use of a 360 video to help patients be informed. This is an interesting study and relevant, but I do have several suggestions outlined below that would improve the quality of the findings. Also, I agree with the editor that they should provide the translation procedure for the English quotes and have a native speaker review the revised manuscript.

Abstract

- I feel that the abstract is missing the actual conclusions. The results are stated but the implications of the findings are not clear.

Introduction

- Suggest review for minor typos (extra spaces, missing commas, “postpone” does not need to be hyphenated)

- The items in this sentence don't seem to be mutually exclusive: "However, it is not always clear what information patients want, as some prefer wide-ranging information (4, 5), whilst others prefer detailed information (4, 6)." Perhaps by wide-ranging, you mean more general information?

- For this sentence, “Currently, patients have access to the Internet, which offers unlimited amounts of health information(7).” Suggest changing ‘currently’ to ‘most’ – not all patients have internet access.

- I’m not sure how I feel about this sentence: “In our hospital we had the impression that patients were missing certain information and that they did not read or (fully) understand the provided (written) information (personal opinion).”

- The entire paper is about the provision of a virtual reality tool but there is only one sentence about VR in the introduction. This seems off – I would suggest adding more on the evidence and introduction of this concept.

Methods

- The authors state the used COREQ however they did not include a copy of the completed checklist as a part of their submission. Perhaps not a requirement for the journal but it strengthens the submission.

- Most methods section is written in the passive voice. This is a choice but the manuscript would be much more acceptable to a lay reader if it was written in the active voice.

- Most of the information provided under the subheading ‘Participants’ is really material that should be reserved for the results section. This includes all of the information starting with the sentence “In total, 21 patients were contacted…” through the end of this subsection. Along with the information in the next section on the length of the interviews.

- I think there is a word missing from this sentence: “This 360 ° video offered the patients to virtually experience some parts of the operation day and to see some real operation scenes of a laparoscopic inguinal hernia repair procedure.”

- Given that the themes in the results include in-person requests, I would suggest adding the dates of these interviews due to the COVID-19 shifts that you acknowledge in the conclusions.

- I see no mention of data saturation which is one of the COREQ checklist items and is important in qualitative research. The authors mention that they started coding before interviewing was complete so perhaps this was in fact done?

Results

- I’m not following the connection between the subheading ‘visual tools’ and the content under it – I don’t see references to visual tools here?

- The authors seem to have done a nice job responding to previous reviewers comments about breaking up the quotes and improving the clarity of the themes.

- I would highly suggest a table to layout the major themes from the analysis. With only one table in the manuscript at present, this could really strengthen the summarization of the findings.

- I’m don’t think the word ‘sparring’ is used correctly in this sentence: “For others, the Internet was not perceived as helpful in providing sufficient confidence for an active role as a sparring partner.”

- You mention in the abstract that the patients did not find the VR fearsome but I don’t see this as a theme in the results. Perhaps I missed it? I was sort of looking for a heading on the acceptability of the VR but perhaps not enough information emerged on this?

Discussion

- You say “these studies” but then only one citation, which I see is a systematic review. Perhaps you could say instead introduce citation 14 as a 2018 systematic review and then in the next sentence say “the studies in this review were mainly conducted…”

- This sentence sits at the end of a paragraph without further explanation: “Alternative options could be for example use of autobots.” I feel this contextual paragraph needs improvement. Surely there are other results in the literature that indicate if your findings are in agreement with other studies that patients felt there was a lack of tailored information?

- When I read sentences like: “The majority of the patients believed that the 360 ° video enhanced their understanding, especially with regard to the surgical procedural information.” In the discussion, I feel like this is missing from the results. Perhaps I missed it but again I do wonder if a section about the acceptability of the 360 tool would be appropriate in the results section.

- In the section under “seeing is understanding” – is there other research around VR that could be incorporated so that you are reviewing your results against the current literature on VR?

- I think you can remove “for instance” from the last line of the manuscript.

7. PLOS authors have the option to publish the peer review history of their article (what does this mean?). If published, this will include your full peer review and any attached files.

Reviewer #2: No

---

## [Author Response · Author response to Decision Letter 1]

25 Sep 2020

Abstract

- I feel that the abstract is missing the actual conclusions. The results are stated but the implications of the findings are not clear.

We changed the abstract and make a division between the results and the conclusion.

Conclusion: In order to address to patients’ information needs, complementary tools or services are needed that increase personal contact as well as tailoring with respect to individual needs. Even though video-apps are a partial alternative, hospitals should still offer patients the possibility for face-to-face meetings with physicians as this is highly valued by patients and for instance leads to increased trust in physicians’ performance.

Introduction

- Suggest review for minor typos (extra spaces, missing commas, “postpone” does not need to be hyphenated)

We reviewed the paper and adapted it accordingly

- The items in this sentence don't seem to be mutually exclusive: "However, it is not always clear what information patients want, as some prefer wide-ranging information (4, 5), whilst others prefer detailed information (4, 6)." Perhaps by wide-ranging, you mean more general information?

We changed it to the following: 

“However, it is not always clear what information patients want as some prefer receiving very general information about just the procedure (4, 5), whilst others prefer detailed information for example on outcomes and survival (4, 6).”

- For this sentence, “Currently, patients have access to the Internet, which offers unlimited amounts of health information(7).” Suggest changing ‘currently’ to ‘most’ – not all patients have internet access.

Changed accordingly 

- I’m not sure how I feel about this sentence: “In our hospital we had the impression that patients were missing certain information and that they did not read or (fully) understand the provided (written) information (personal opinion).”

We removed the word “personal opinion”. We want to indicate that we noted that patients had not read or understood the provided information as they for example had forgotten to take medication prior to the surgery, even though this was written in the instructions. Therefore we wanted to see whether we could improve the understanding of the procedure and the process. 

- The entire paper is about the provision of a virtual reality tool but there is only one sentence about VR in the introduction. This seems off – I would suggest adding more on the evidence and introduction of this concept.

We added more information on the use of VR to the introduction. 

“Virtual reality has been used to train physicians or residents to improve their communication skills towards the patient (11). Furthermore, one study suggested that informing patients on a surgical procedure via VR might improve patient comprehension of their condition (12).”

Methods

- The authors state the used COREQ however they did not include a copy of the completed checklist as a part of their submission. Perhaps not a requirement for the journal but it strengthens the submission.

- Most methods section is written in the passive voice. This is a choice but the manuscript would be much more acceptable to a lay reader if it was written in the active voice.

We made the method section more “active”. 

- Most of the information provided under the subheading ‘Participants’ is really material that should be reserved for the results section. This includes all of the information starting with the sentence “In total, 21 patients were contacted…” through the end of this subsection. Along with the information in the next section on the length of the interviews.

We changed this accordingly 

- I think there is a word missing from this sentence: “This 360 ° video offered the patients to virtually experience some parts of the operation day and to see some real operation scenes of a laparoscopic inguinal hernia repair procedure.”

We changed the sentence

“In this 360˚ video patients could virtually experience parts of the day of the surgery and see parts of a laparoscopic inguinal hernia repair procedure.”

- Given that the themes in the results include in-person requests, I would suggest adding the dates of these interviews due to the COVID-19 shifts that you acknowledge in the conclusions.

We added the period to the methods section (study design)

Interviews were held between September and December 2018. 

- I see no mention of data saturation which is one of the COREQ checklist items and is important in qualitative research. The authors mention that they started coding before interviewing was complete so perhaps this was in fact done?

We added the COREQ checklist and added information on when the data collection was ended (methods – data collection)

The data collection was ended after saturation was reached.

Results

- I’m not following the connection between the subheading ‘visual tools’ and the content under it – I 

don’t see references to visual tools here?

We changed the subchapter title to Using tools

- The authors seem to have done a nice job responding to previous reviewers comments about breaking up the quotes and improving the clarity of the themes.

We thank the reviewer for his/her nice comments – we believe that the comments of the reviewer strongly improved our paper 

- I would highly suggest a table to layout the major themes from the analysis. With only one table in the manuscript at present, this could really strengthen the summarization of the findings.

We added a table 2. 

- I’m don’t think the word ‘sparring’ is used correctly in this sentence: “For others, the Internet was not perceived as helpful in providing sufficient confidence for an active role as a sparring partner.”

We changed the wording to “for an active role in the share d decision making process”

- You mention in the abstract that the patients did not find the VR fearsome but I don’t see this as a theme in the results. Perhaps I missed it? I was sort of looking for a heading on the acceptability of the VR but perhaps not enough information emerged on this?

The VR video was strongly accepted, and not of the patients found it troublesome. There was too little information to have an entire theme on this topic. However, we combined the finding from various places to the end of the results section 

Still, the use of the VR-video was accepted by all patients and none of the patients found it fearsome themselves. Beforehand we had expected that patients might find it fearsome to view a surgical procedure in 360. Some of the patients indicated that it could scare people off to get so close to the real surgery, although they did not find it problematic themselves. In addition there was one patient who indicated not to be interested in the video and who had the interview by phone.  

Discussion

- You say “these studies” but then only one citation, which I see is a systematic review. Perhaps you could say instead introduce citation 14 as a 2018 systematic review and then in the next sentence say “the studies in this review were mainly conducted…” 

We changed the wording

However, most of the studies included in this review were conducted before 2010.

- This sentence sits at the end of a paragraph without further explanation: “Alternative options could be for example use of autobots.” 

We added some literature on this.

Alternative options could be, for example, use of chatbots. For the latter, studies in breast cancer patients, for example, have shown that an artificial conversation agent achieved equal satisfaction scores as “real” physicians (18).

I feel this contextual paragraph needs improvement. Surely there are other results in the literature that indicate if your findings are in agreement with other studies that patients felt there was a lack of tailored information?

We agree and added some more information and results from the literature

Patients felt that the general information obtained from the brochures, the internet or doctors assistants, was not sufficiently tailored to their personal situation, for instance for questions regarding their recovery process. It had been shown previously that tailored information provided on the internet leads to higher satisfaction rates among patients, especially among younger patients, although no improvements in patients outcomes have been found (17,18). Some of our patients had WhatsApp contact with their surgeon, which was highly appreciated. However, this would be nearly impossible to provide to all patients, as physicians are often afraid of being overloaded with questions from patients. Still research has shown benefits of a short messaging system protocol leading to a reduced number of clinic visits and increased efficiency of clinic visits(19). Alternative options could be for example use of chatbots. For the latter, studies in breast cancer patients, for example, have shown that an artificial conversation agent achieved equal satisfaction scores as “real” physicians (20). 

- When I read sentences like: “The majority of the patients believed that the 360 ° video enhanced their understanding, especially with regard to the surgical procedural information.” In the discussion, I feel like this is missing from the results. Perhaps I missed it but again I do wonder if a section about the acceptability of the 360 tool would be appropriate in the results section.

See above. 

- In the section under “seeing is understanding” – is there other research around VR that could be incorporated so that you are reviewing your results against the current literature on VR?

We added some literature

Also in other fields it has been suggested that VR can help in pharmacy education by providing immersive educational activities (24), having elderly patients coping with dementia (25) or teaching skills to health professionals (26).

- I think you can remove “for instance” from the last line of the manuscript.

- Adapted accordingly

---

## [Editor Report · Decision Letter 2]

28 Sep 2020

Patients want to be seen: the top 3 information needs of patients with inguinal hernia.

PONE-D-19-28443R2

Dear Dr. van Stralen,

We’re pleased to inform you that your manuscript has been judged scientifically suitable for publication and will be formally accepted for publication once it meets all outstanding technical requirements.

Kind regards,

M Barton Laws

Academic Editor

PLOS ONE

Additional Editor Comments (optional):

While I have recommended acceptance, I urge you to proofread the paper carefully before publication. In future submissions, you should always make sure to number the pages, as this is helpful to editors and reviewers. Because you have not done so it is difficult for me to point out specific issues. For example, in the first paragraph of the results there is a typographical error of "latest" for "lasted." You might also try to have a native English speaker review this as some of the wording is not really standard, though it is certainly comprehensible. Nevertheless this is a useful contribution that may inform practice.
---

## [Editor Report · Acceptance letter]

2 Oct 2020

PONE-D-19-28443R2 

Patients want to be seen: the top 3 information needs of patients with inguinal hernia. 

Dear Dr. van Stralen:

I'm pleased to inform you that your manuscript has been deemed suitable for publication in PLOS ONE. Congratulations! Your manuscript is now with our production department. 

Kind regards, 

on behalf of

Dr. M Barton Laws 

Academic Editor

PLOS ONE